# The Promotive and Inhibitory Role of Long Non-Coding RNAs in Endometrial Cancer Course—A Review

**DOI:** 10.3390/cancers16112125

**Published:** 2024-06-03

**Authors:** Patryk Jasielski, Izabela Zawlik, Anna Bogaczyk, Natalia Potocka, Sylwia Paszek, Michał Maźniak, Aleksandra Witkoś, Adrianna Korzystka, Aleksandra Kmieć, Tomasz Kluz

**Affiliations:** 1Department of Gynecology, Gynecology Oncology and Obstetrics, Fryderyk Chopin University Hospital, 35-055 Rzeszow, Poland; 2Laboratory of Molecular Biology, Centre for Innovative Research in Medical and Natural Sciences, Medical College of Rzeszow University, 35-959 Rzeszow, Poland; 3Institute of Medical Sciences, Medical College of Rzeszow University, 35-959 Rzeszow, Poland

**Keywords:** endometrial cancer, lncRNA, long non-coding RNA, carcinogenesis

## Abstract

**Simple Summary:**

Endometrial cancer has emerged as the main gynecological malignant tumour in developed countries. The development of molecular techniques allows the precise profiling of each case. This affects the selection of a therapy and the patient’s prognosis. Long non-coding RNAs, which influence tumour development, are a novelty in endometrial cancer. They can both stimulate and inhibit the progression of the disease. Their number is constantly growing, and the mechanisms of action are not fully discovered. Some of them may become targets for new molecular therapies in the future.

**Abstract:**

Endometrial cancer is one of the most common malignant tumours in women. The development of this tumour is associated with several genetic disorders, many of which are still unknown. One type of RNA molecules currently being intensively studied in many types of cancer are long non-coding RNAs (lncRNAs). LncRNA-coding genes occupy a large fraction of the human genome. LncRNAs regulate many aspects of cell development, metabolism, and other physiological processes. Diverse types of lncRNA can function as a tumour suppressor or an oncogene that can alter migration, invasion, cell proliferation, apoptosis, and immune system response. Recent studies suggest that selected lncRNAs are important in an endometrial cancer course. Our article describes over 70 lncRNAs involved in the development of endometrial cancer, which were studied via in vivo and in vitro research. It was proved that lncRNAs could both promote and inhibit the development of endometrial cancer. In the future, lncRNAs may become an important therapeutic target. The aim of this study is to review the role of lncRNAs in the development of carcinoma of uterine body.

## 1. Introduction

### 1.1. Endometrial Cancer

Endometrial cancer (EC) is considered one of the most common malignant tumours of the female reproductive system. In 2020, 417,367 new cases of this cancer were diagnosed worldwide. A total of 97,370 women died, and still the incidence of EC continues increasing, especially in North America and Western Europe [1]. This is related to the ageing of the population, but also to the increasing rate of obesity, which is the main risk factor for this cancer as it is responsible for over 50% of cases [2]. Other risk factors include smoking, type 2 diabetes, polycystic ovarian syndrome, or estrogen therapy after menopause [3]. EC can be divided into morphological and molecular classifications. Historically, Bokhman (1983) suggested the first subtype classification of endometrial carcinoma, which distinguished between type I and type II endometrial cancer. The classification was based on clinical and hormonal traits. Type I, called endometrioid endometrial cancer, includes grade I-II endometroid adenocarcinoma. It accounts for 80% of all EC and is correlated with an excessive estrogen stimulation. Type II, called non-endometrioid endometrial cancer, is less common, and has a worse prognosis than type I. Type II EC is divided into grade III endometrioid adenocarcinoma, serous clear cell, undifferentiated, and carcinosarcomas [4,5]. The Cancer Genome Atlas presented the molecular classification of EC. It consists of four subgroups: POLE (ultra-mutated), microsatellite instability hypermutated or mismatch repair deficiency microsatellite instable (MSI), copy-number low (microsatellite stable), and copy-number high or p53 abnormal (serous-like) [6,7]. Each subgroup is correlated with different genes mutations and prognosis. EC is treated by hysterectomy with pelvic lymph nodes excision. Depending on the stage of cancer, chemotherapy, radiotherapy, and chemoradiotherapy can also be deployed [3].

### 1.2. Long Non-Coding RNAs (lncRNAs)

Non-coding RNAs are molecules which are not translated into proteins. However, this definition is not fully correct. Recent research has shown that some long non-coding RNAs (lncRNAs) encode the formation of small peptide molecules. LncRNAs can influence cell metabolism and tissue functioning in various ways, not only in connection with proteins synthesis. According to their length, they can be divided into two groups: short non-coding RNA and long non-coding RNA. It was established that lncRNA is a transcripted particle of RNA that is more than 200 nucleotides in length and is not translated into proteins. According to the origin (genomic origin), there are differentiated subtypes of lncRNA: intronic lncRNA, intergenic lncRNA, antisense lncRNA, bidirectional lncRNA, enhancer RNA, and circular lncRNA (Table 1) [8,9,10].

LncRNA is a product of polymerase RNA. Hence, the biogenesis of this molecule is like messenger RNA (mRNA). Due to polyadenylation, alternative splicing, and cleavage different isoforms of lncRNA can be created. Additionally, LncRNAs can be modified by some transcription factors like p53, Sox2, or nuclear factor-kappa B (NF-κB) [11,12]. LncRNA can be found in the nucleus, nucleolus, cytoplasm, and mitochondria. Currently, many lncRNA molecules are being discovered in the human body, and their number is constantly growing. Even though they are not responsible for the formation of proteins, they perform several crucial functions in the body. LncRNA takes part in the regulation of the genome, cell structure, and gene expression. It regulates the chromatin structure, transcription, translation, and processing of RNA. LncRNAs are also involved in the development of cells, the regulation of the immune system, the expression of cytokines, and the maintenance of homeostasis [13,14,15]. On the other hand, lncRNAs have an impact on pathological processes. They influence the development of the cardiovascular system, as well as cardiovascular diseases such as hypertension, heart failure, and coronary artery disease [16]. Moreover, it has been proved that lncRNAs participate in pathogenesis of certain neurological conditions such as Alzheimer’s disease, Parkinson’s disease, and lateral spinal cord sclerosis. It is correlated with the role of lncRNAs in synapse formation and neural cell functioning [17]. LncRNAs are also crucial factors in cancer development, the course of the disease and efficiency of treatment [18].

Due to the growing number of endometrial cancer cases, the search for individualized therapy that is less burdensome for patients is required. LncRNAs have broad and incompletely described effects on a number of metabolic processes. Due to the lack of a similar article, the topic of our review is the comprehensive description of the stimulatory and inhibitory effects of lncRNAs on the course of endometrial cancer. We took into account the effect of lncRNA both promoting and inhibiting the progression of endometrial cancer because in the future there is a chance to use drugs with different mechanisms of action.

## 2. Materials and Methods

A search for articles about the lncRNA in endometrial cancer was performed. The database of Pubmed was analyzed. The search was undertaken in April 2024. To find articles, the following keywords were used: “endometrial cancer”, “lncRNA”, ”long non-coding RNA”, “cancer of the uterine body”. A three-step analysis of found articles—title, abstract, and entire text—was undertaken. The inclusion criteria were as follows: (1) articles from the last 10 years; (2) manuscripts written in English; (3) original articles; (4) articles with access to the entire text; (5) studies on cells, animals, and humans.

## 3. LncRNA Mechanisms of Action

The action of lncRNAs is related to their influence on gene expression. The mechanisms of this action are complicated and still not fully understood. LncRNAs can regulate gene expression at multiple levels by interacting with RNA, DNA, and proteins. LncRNA mechanisms of action can be divided into three main groups: (1) regulation of gene expression at the transcription level by binding to DNA or transcription factors; (2) post-transcriptional action by interfering in the activities of mRNAs, miRNAs, or proteins; (3) epigenetic influence—activation or repression of gene expression by impact on chromatin complexes. LncRNAs have an ability to create hybrid structures with DNA (triple helices). It can change accessibility to chromatin and mediate gene silencing or activation [19,20,21]. The activation of proto-oncogene sphingosine kinase 1 (SPHK1) by lncRNA KHPS1 is an example of such a mechanism This molecule creates a triple helix with the SPHK1 enhancer to activate the transcription of SPHK1. Experimentally, KHPS1 was replaced by other lncRNA-MEG3. The correlation between SPHK1 and MEG3 did not activate this proto-oncogene [22].

LncRNAs are important gene expression regulators. They are located between genes, where they can have an impact on transcription. Antisense and bidirectional lncRNAs are particularly involved in these processes. The lncRNA transcript can regulate adjacent gene expression. Moreover, the transcription or splicing of the lncRNA can influence chromatin state and therethrough the expression of adjacent genes. LncRNAs, on the other hand, can suppress or activate polymerase II promotors, as well as change histone structure. Additionally, they can modify the enhancers of genes to activate or block expression [23,24,25,26]. LncRNAs can also function as post-transcriptional, translational, and post-translational regulators. Some lncRNAs are translated into peptides [27]. Long non-coding RNA can regulate protein functions by binding to specific RNA sequences and creating lncRNA–protein complexes. Another mechanism is to modify the splicing of pre-mRNA. It happens through alteration of splicing factors and splicing blocking by establishing connections with pre-mRNA. Moreover, LncRNAs can also make connections with structures that interact with certain proteins. This may cause the blocking or enhancement of the action [28,29,30]. LncRNAs can bind with other RNAs and activate proteins engaged in mRNA degeneration [31].

Finally, lncRNA has an impact on proteins located in the neighbourhood of chromatin where it can regulate DNA expression. These molecules are also connected with nuclear condensates—compartments engaged in cell functions [32]. On the other hand, lncRNA is a scaffolding particle. This means that this RNA creates architectural scaffolds for RNA and protein interactions in the nucleus and in the cytoplasm [33].

Some lncRNAs have special complementary sites to microRNAs (miRNAs) which, in turn, participate in the regulation of mRNA expression. Thanks to complementary fragments, lncRNA molecules combine with miRNA, which reduces the pool of free molecules necessary for the functioning of mRNA. This affects the regulation of all cellular processes [34].

In summary, the above description provides only a brief explanation of the mechanisms of lncRNAs’ action. The issue itself is broad and the mechanisms of action of lncRNAs are not yet fully understood. The functions of lncRNAs in the body are summarized in Table 2.

## 4. The Role of lncRNA in Endometrial Cancer

As mentioned earlier, lncRNAs influence the occurrence and the course of carcinogenesis. Individual long non-coding RNA molecules are associated with various cancers. Below there are described general characteristics of certain lncRNA molecules and their impact on the development, course, and treatment of endometrial cancer.

### 4.1. Long Non-Coding Antisense RNAs (-AS) Group

Antisense RNAs are the largest group of lncRNAs. It is a natural antisense transcript which contains single-stranded RNA that is complementary to messenger RNA (mRNA)—a protein-coding molecule. Antisense lncRNAs can have an impact on gene expression by blocking the translation of mRNA into proteins. Depending on their mechanism of action, antisense lncRNAs can stimulate or inhibit the development of endometrial cancer. The first example of an antisense lncRNA is NR2F1-AS1. It promotes the progression of endometrial cancer. This antisense lncRNA stimulates the migration and invasion of EC cells due to a decrease in microRNA-363 (miR-363) expression through the phosphoinositide 3-kinases/serine/threonine protein kinase/glycogen synthase kinase-3β (PI3K/AKT/GSK-3β) pathway. MiR-363 inhibits the expression of the SRY-related HMG-box 4 (SOX4) oncogene, which promotes tumour growth and metastasis [35]. In another study, LOXL1 antisense lncRNA has a similar impact on EC as it promotes the invasion and progression of cancer by downregulation expression of miR-28-5p. This microRNA decreases the expression of Ras-related protein Rap-1b gene (RAP1B), which is a part of the oncogene family. RAP1B stimulates the development of endometrial cancer [36]. MCM3AP-AS1 is another antisense lncRNA promoting EC development. This molecule stimulates the expression of the vascular endothelial growth factor (VEGF) gene. It engages in the formation of new blood vessels, which increases the blood supply to the tumour and promotes its development and metastasis. MCM3AP-AS1 works by blocking miR-126, which in turn blocks VEGF expression [37]. It has been shown that AFAP1-AS1 contributes to the progression of EC by promoting vascular endothelial growth factor A (VEGFA) expression. This gene increases the angiogenesis of tumour vessels. AFAP1-AS1 blocks miR-545-3p from acting [38]. NNT-AS1 is another antisense lncRNA that boosts the development of endometrial cancer. This lncRNA inhibits the expression of miR-30c-microRNA, which blocks EC progression. NNT-AS promotes further growth of estrogen-mediated uterine corpus cancer [39]. Interestingly, DLX6 antisense-1 stimulates the proliferation and invasion of endometrial cancer cells, simultaneously impeding apoptosis. It was proved that silencing of this lncRNA repressed the proliferation and invasion of EC cells. It is regulated by the interaction with p300/E2F1 acetyltransferase [40]. TTN-AS1 promotes uterine corpus endometrial cancer development. This molecule promotes the expression of pumilio homolog 2 (PUM2), which is a translational regulator. TTN-AS1 blocks miR-376a-3p expression, which is the factor that suppresses EC progression [41]. Moreover, GATA3-AS1 stimulates the invasion and migration of EC cells by blocking miR-361, and promotes the activation of Arrestin beta 2 gene (ARRB2), which, among others, is involved in cecum lymphoma occurrence. The GATA3-AS1/miR-361/ARRB2 axis was shown to regulate EC cell proliferation, invasion, and migration [42]. In another study, it was demonstrated that HOXB-AS3 also stimulates uterine corpus endometrial cancer by sponging miR-498-5p. This microRNA downregulates the expression of disintegrin and metalloproteinase domain-9 (ADAM-9). This enzyme level is increased in different cancers including EC [43]. Furthermore, HOXB-AS3 has an impact on lipid metabolism in EC by binding to polypyrimidine tract-binding protein 1 (PTBP1). It promotes supply of energy to growing tumour [44]. Another antisense lncRNA HOXA-AS2 promotes the invasion and proliferation of endometrial cancer type 1 by silencing miRNA-302c-3p and promoting the expression of zinc finger X-chromosomal protein (ZFX). This protein stimulates a growth and migration of cancer cells [45]. The next lncRNA HOXC-AS2 advertises EC development by blocking miR-876-5p. It decreases the expression of hexokinase domain-containing 1 (HKDC1). This enzyme provides a supply of glucose for tumour development, and promotes the production of reactive oxygen species. These forms of oxygen have an impact on pyroptosis, which is a form of programmed cell death and which plays a crucial role in a wide range of diseases [46]. The FIGO stage, histological grade, and lymph node metastasis are correlated with another long non-coding RNA-RHPN1-AS1. It reacts with mitogen-activated protein kinases (MAPKs) and extracellular signal-regulated kinases (ERKs)—the MAPK/ERK pathway [47]. Another mechanism of antisense lncRNA action is up-regulating cyclin D1, a protein that alters the cell cycle and promotes tumorigenesis. The regulation of cyclin D1 is associated with the presence of lncRNA ABHD11-AS1 [48]. LncRNA DCST1-AS1 enhances tumour growth, migration, and invasion. This lncRNA induces the expression of cell adhesion molecule 1 (CADM1) and homeobox B5 (HOXB5) by miR-873-5p and miR-665. Increased expression of HOXB5 and CADM1 is correlated with tumour development [49]. The overexpression of another lncRNA-DSCAM-AS1 is associated with higher expression levels of the prolactin gene (PRL) and estrogen receptor α (Erα), a more aggressive course of EC, and a shorter survival period of patients. The oncogenic role of DSCAM-AS1 is correlated with the blocking of miR-136-5p actions [50,51]. In other research, it has been shown that VPS9D1-AS1 can contribute to EC progression by binding to miR-520a-5p and promoting the expression of Baculoviral IAP Repeat-Containing 5 (BIRC5). The protein coded by this gene inhibits the apoptosis of endometrial cancer cells. VPS9D1-AS1 regulates the expression by sponging miR-377-3p and stimulates the glucocorticoid-regulated kinase 1 (SGK1) activity. It correlates with lymph node metastasis and the FIGO stage. SGK1 is engaged in cellular stress response and cell survival [52,53]. Other examples of antisense lncRNAs that promote EC development are BMPR1B-AS1, which sponges miR-7-2-3p and modulates the DCLK1/Akt/NF-κB pathway, and ZFAS. This lncRNA promotes higher expression of cyclin D1, ZNRD1-AS1 and NIFK-AS1 [54,55,56]. It has been shown that antisense lncRNAs can promote EC development by reducing the effectiveness of chemotherapy., whereas LncRNA TMPO-AS1 promotes glucose transporter-1 (GLUT1)-dependent glycolysis and resistance to paclitaxel therapies. Paclitaxel is one of the cytostatics that is used in the treatment of different cancers including endometrial cancer. GLUT1 stimulates the proliferation and invasion of endometrial cancer cells. TMPO-AS1 elevated GLUT1 expression by directly binding to two tumour suppressors, microRNAs-miR-140 and miR-143. The downregulation of TMPO-AS1 significantly inhibited EC proliferation and reduced paclitaxel resistance [57]. Moreover, CDKN2B-AS1 has a similar effect and it increases resistance to paclitaxel by blocking the expression of miR-125a-5p [58]. The promotion of endometrial cancer cell proliferation and its migration and paclitaxel resistance is correlated with the higher expression of another antisense lncRNA, FGD5-AS1. Moreover, FGD5-AS1 impedes immune escape via the PD-1/PD-L1 checkpoint [59].

Some antisense lncRNAs have inhibitory effects on the development of endometrial cancer. This is related to various molecular mechanisms. The first example is FRMD6-AS2. This lncRNA suppresses tumour growth, migration, and invasion by the activation of Hippo signalling. The Hippo pathway regulates cell proliferation, and its activation suppresses this process. FRMD6-AS2 has also been shown to have an impact on the function of contraction and actin-mediated cell movement [60]. It has been shown that MTCP1-AS1 is another lncRNA that has a negative impact on EC cell proliferation, migration, invasion, and epithelial to mesenchymal transition. This lncRNA inhibits miR-650 and promotes the expression of mothers against decapentaplegic homolog 7 (SMAD-7). It is a member of the group which is engaged in leukocytes signals transfer to cells. SMADs are correlated with tumour growth suppression [61]. LncRNA can also inhibit EC progression by stimulating the apoptosis of endometrial cancer cells. The example is EIF1AX-AS1. The mechanism of this action is that EIF1AX-AS1 mediates the eukaryotic translation initiation factor 1A X-linked (EIF1AX) mRNA instability by binding to poly C binding protein 1 (PCBP1). The instability of this mRNA results in the dysfunction of EC cells [62]. Moreover, lncRNA can increase the degradation of Aurora kinase A (AURKA), which is involved in cell proliferation. This mechanism is related to lncRNA SOCS2-AS1. This molecule suppresses EC cell proliferation and promotes cell-cycle arrest, as well as apoptosis [63,64]. In another study, it was proved that GATA6-AS inhibited EC progression by downregulation of metalloproteinase 9 (MMP-9) expression. This protein promotes metastasis and angiogenesis due to the decomposition of the extracellular matrix [65]. In the study by Liu et al., it was proved that OIP5-AS1 inhibited EC progression by blocking miR-200c-3p and controlled the phosphatase and tensin homolog (PTEN/AKT) pathway. PTEN is a tumour suppressor gene and participates in cell cycle, angiogenesis, migration, and invasion [66]. LncRNA inhibits the migration and invasion of EC cells by decreasing expression of neuromedin U (NMU). It is proved that this neuropeptide promotes endometrial cancer development. This is how another lncRNA HAND2-AS1 works [67]. Apart from these all, antisense lncRNAs can also mitigate EC progression by the stimulation of the immune system activity. They may exert such an effect through various mechanisms, including stimulating the activity of macrophages and other leukocytes, signalling pathways, or cytokine secretion [68,69]. The main characteristics of long non-coding antisense RNAs are summarized in Table 3.

### 4.2. Long Intergenic Non-Coding RNA (LINC)

Long intergenic non-coding RNA is one kind of lncRNA involved in various cellular functions, including the regulation of gene expression and chromatin remodelling. The abnormal expression of lincRNAs can induce or suppress EC development.

One of the lincs that is important in the development of EC is linc02936. It promotes EC progression by inhibiting ferroptosis—the process of programmed cell death dependent on iron and mediated by enzyme ceruloplasmin. Linc02936 upregulates the expression of ceruloplasmin by binding to SIX Homeobox 1 (SIX1) [70]. Another lncRNA is linc01016. It stimulates EC progression by inhibiting miR-302a-3p and miR-3130-3p. These microRNAs block the expression of nuclear factor YA (NFYA), which promotes endometrial cancer development [71]. Furthermore, another molecule, linc01410, stimulates the expression of chromodomain helicase DNA-binding protein 7 (CHD7) and inhibits miR-23c. This protein engages in promoting the angiogenesis of the tumour. CHD7 is impeded by miR-23c. Hence, linc01410 promotes EC progression [72]. In another study, it was proved that linc01857 was engaged in the EC course as well. The decreased expression of this lncRNA promotes EC cell apoptosis and blocks proliferation and migration. It stabilizes proto-oncogene MYCN, as well as inhibits miR-19b-3p expression, which has a negative impact on MYCN expression [73]. Moreover, lincs can promote EC progression by binding to MYC and deactivation of PTEN. This is a potential mechanism of action of inc00470 [74]. Apart from all this, linc01194 stimulates EC progression by connecting to insulin-like growth factor 2 binding protein 1 (IGF2BP1). This factor promotes the expression of sex-determining region Y-box 2 (SOX2). This gene increases tumour development and promotes metastasis occurrence [75]. Linc can either promote endometroid cancer proliferation by stimulating overexpression of VEGFA, and thus increases the angiogenesis of tumour vessels. It is correlated with linc01541 [76]. Linc00958 has a comprehensive impact on EC development. This lncRNA promotes insulin-like growth factor mRNA-binding protein 3 (IGF2BP3). This protein, in turn, upregulates the expression of E2F3—a gene that increases EC cell proliferation and tumour growth. Moreover, linc00958 can upregulate PHD Finger Protein 6 (PHF6) by impeding miR-3174 activity. PHF6 has an oncogenic function in EC. Linc00958 may also inhibit miR-145-3p expression, and as a result, it leads to an increase in transcription factor 4 (TCF-4) activity, which causes EC progression [77,78,79]. Furthermore, linc01224 and linc01170 bind to and downregulate miR-485-5p. This results in an increase in threonine-protein kinase (AKT3) expression, which leads to EC progression and metastasis [80,81]. In another study, linc01220 generates the overexpression of mitogen-activated protein kinase 11 (MAPK11). This kinase increases the proliferation of EC cells and stimulates the progression of the disease [82]. Another oncogenic long intergenic non-coding RNA is linc00461. This lincRNA upregulates the expression of cyclooxygenase-2 (COX-2) by decreasing the expression of miR-219-5p, for which COX-2 is a target. COX-2 promotes tumour growth and EC progression, as well as decreases its immune system activity [83]. The last lincRNA to mention that promotes EC progression is linc01106. This molecule inhibits the activity of miR-449a and upregulates the expression of proto-oncogene MET, which has a negative impact on EC course [84].

On the other hand, there is a group of lincRNAs that inhibit the development of endometrial cancer. Linc00672 suppresses the progression of EC by two mechanisms. Firstly, this lncRNA promotes the p53-induced downregulation of LIM and SH3 Protein 1 (LASP1). This protein induces cytoskeleton alteration, the progression of cancer, and metastasis. Secondly, ling00672 increases chemosensitivity to paclitaxel [85]. Forkhead box protein O1 (FOXO1) inhibits endometrial cancer growth by suppressing angiogenesis and promoting apoptosis. FOXO1 is upregulated by linc00261 and downregulated by miR-182, whose expression is impeded by linc00261 [86]. Polypyrimidine tract binding protein 1 (PTBP1) has a negative impact on prognosis of EC by promoting the formation of pseudopodia, which are engaged in the invasion and occurrence of metastasis. Linc00478 inhibits EC progression by downregulating PTBP1 expression [87]. The last linc to mention here is linc01589. It suppresses EC development by the stimulation of immune system activation due to the increasing expression of B cells, T cells, and NK cells [88]. The main characteristics of long intergenic non-coding RNA are summarized in Table 4.

### 4.3. Maternally Expressed Gene 3 (MEG3)

There is also a group of specific lncRNAs which are of particular interest to scientists. They have an impact on the course of different cancers. The first example, maternally expressed gene 3 (MEG3), is an intergenic lncRNA that participates in the course of endometrial cancer. It is expressed in a wide range of tissues and regulates gene activity. MEG3 is considered as a tumour suppressor in cancers including EC. MEG3 is downregulated in uterine corpus endometrial cancer cells. Its effect in blocking EC development may be related to several mechanisms. MEG3 increases the expression of programmed death-ligand-1 (PD-L1) which acts as a tumour suppressor, and its levels are downregulated in EC cells. PDL-L1 downregulation is correlated with an aggressive course of cancer and poor prognosis. PD-L1 has an anti-tumour activity through inhibiting myeloid cell leukemia-1 (MCL1) expression. MCL1 promotes cell survival and the epithelial–mesenchymal transition. This process induces invasion and promotes metastasis occurrence. PD-L1 expression is downregulated by miR-216a. MEG3 stimulates PD-L1 expression by inhibiting activation of miR-216a. Hence, MEG3 suppresses cells migration and invasion [89]. Another potential mechanism of MEG3 action is the downregulating activity of phosphoinositide 3-kinases/mammalian target of rapamycin (PI3K/m-TOR) signalling. PI3K/m-TOR dysfunction is correlated with carcinogenesis—it promotes progression and invasion, and inhibits apoptosis [90]. Moreover, MEG3 can downregulate the activity of neurogenic locus notch homolog protein 1 (Notch 1) and hairy and enhancer of split-1 (HES-1). Both proteins are engaged in cancer progression by promoting EC cell proliferation and metastasis, and inhibiting apoptosis [91]. Interestingly, lncRNAs can influence each other’s expression. An example is the downregulation of MEG3 expression by another lncRNA, PSMG3-AS1. MEG3 inhibits EC progression, while PSMG3-AS1 promotes the invasion of this cancer [92].

### 4.4. Homeobox (HOX) Transcript Antisense Intergenic RNA (HOTAIR)

Another well-known lncRNA molecule, the role of which was investigated in several types of cancers including breast, lung, and liver, is homeobox (HOX) transcript antisense intergenic RNA (HOTAIR). It can regulate the expression of homeobox cluster genes. HOTAIR is upregulated in a wide range of malignancies, and it is correlated with overall survival and progression-free survival. In research conducted by He et al., HOTAIR expression was higher among patients with EC than in the control group. The expression of HOTAIR was correlated with EC grade and lymph node metastasis. It was also correlated with a depth of myometrial invasion and lymphovascular space invasion. Moreover, the higher expression of HOTAIR indicated a poorer overall survival ratio [93]. HOTAIR expression can be upregulated by estrogen, which promotes EC cell migration. Thus, the estrogen–HOTAIR axis is a potential target in EC therapy [94]. In other research, it was proved that HOTAIR expression was higher in G3 EC than in G1/G2 [95]. Similar actions exhibit long non-coding antisense RNA myeloid-specific 1 (HOTAIRM1). In EC, HOTAIRM1 expression is correlated with the FIGO stage and lymph node metastasis. The downregulation of this lncRNA expression leads to the inhibition of cell proliferation, migration, and invasion and to the epithelial–mesenchymal transition [96]. The HOTAIR mechanism of action is still unclear. However, in EC, it acts as an oncogene by the activation of PI3K/Akt pathway and suppression of PTEN expression [97]. Estrogen-induced HOTAIR expression can be regulated by a variety of factors. One example is miR-646. This microRNA inhibits the expression of HOTAIR in EC [98]. In the research by Łuczak et al., it was proved that there were no differences in HOTAIR expression between different subtypes of EC. A higher expression of HOTAIR was correlated with a shorter overall survival [99]. The overexpression of HOTAIR in EC may also affect a conservative treatment with progesterone. In a study by Chi et al., HOTAIR expression has been shown to be inversely correlated with progesterone receptor B (PRB) expression. The knockdown of HOTAIR can promote medroxyprogesterone sensitivity by upregulating PRB. Hence, the downregulation of HOTAIR can be a potential therapeutic target to overcoming progesterone resistance [100]. The downregulation of HOTAIR by lentivirus-mediated RNA interference in EC cells was correlated with significant suppression of migration, invasion, and cell proliferation. Moreover, it caused cell cycle arrest at the G0/G1 phase. A decline in HOTAIR expression resulted with a decrease in tumorigenesis [101].

### 4.5. H19

Another lncRNA is H19. It is an lncRNA located in 11 chromosomes in the neighbourhood with the insulin-like growth factor II (IGF-2) gene. It is one of the major oncogenic genes and is engaged in the development of a wide range of tumours, including endometrial cancer. H19 participates in carcinogenesis at various stages of disease development and through various mechanisms. According to Lottin et al., H19 is expressed in stromal and myometrial cells of uterus. A high expression is observed in the vicinity of malignant epithelial cells. The expression is correlated with the invasion of cancer cells into the myometrium and disease progression in general [102]. It may be a result of promoting the epithelial–mesenchymal transition by H19 [103]. Moreover, it can promote homeobox A10 (HOXA10) expression by inhibiting miR-612. HOXA10 is engaged in tumour growth and metastasis occurrence, as well as mediating the epithelial–mesenchymal transition. MiR-612 impedes EC progression by blocking HOXA10 expression [104]. Interestingly, H19 levels in EC can be reduced using metformin [105].

### 4.6. Nuclear-Enriched Abundant Transcript 1 (NEAT1)

Another lncRNA that is upregulated in EC and accelerates malignant growth and metastasis is nuclear-enriched abundant transcript 1 (NEAT1). According to existing research, it is proved that NEAT1 is highly expressed in EC cells. A decrease in NEAT1 expression results in the suppression of proliferation, migration, and invasion. NEAT1 downregulates the expression of miR-144-3p, and that leads to the upregulation of Enhancer of zester homolog 2 (EZH2). EZH2 inhibits the transcription of tumour suppressor genes [106]. High NEAT1 expression is correlated with poor prognosis. It can also downregulate miR-361, the particle that is engaged in the suppression of prometastatic gene expression. NEAT1 also promotes the expression of oncogene signal transducer and activator of transcription 3 (STAT3) by blocking miR-26a activity [107,108]. NEAT1 activity can be positively correlated with the FIGO stage and lymph node metastasis. It increases the migration and invasion ability. NEAT1 increases expression of MMP-2, MMP-7, and the insulin-like growth factor 1 (IGF1) [109]. NEAT1 can also suppress miR-214-3p and promote the expression of high-mobility group protein (HMGA1) to regulate the Wnt/β-catenin pathway and stimulate EC progression [110]. Finally, NEAT1 can impede the expression of miR-202-3p and promote T cell immunoglobulin and mucin domain 4 (TIMD4). TIMD4 promotes the progression of EC and reduces the time free from progression [111].

### 4.7. Small Nucleolar Host Genes (SNHG)

Another group of lncRNAs that take part in course of endometrial cancer are small nucleolar host genes (SNHGs). They are a group of lncRNAs that are overexpressed in many cancers. In general, they are involved in tumour development—SNHGs increase the proliferation, cell cycle progression, invasion, and metastasis of cancer cells. SNHGs are created from both introns and exons. They can exert impact on a wide range of cells functions at the DNA/RNA/protein level. The first example of molecule from this group is SNHG9. It promotes EC cell proliferation and glycolysis, processes that generate glucose—a source of energy for tumour development [112]. They can also stimulate the development of EC by upregulating the levels of matrix metalloproteinase MMP2 and MMP9. These enzymes promote invasion and metastasis. This is correlated with SNHG12. SNHG12 expression can be upregulated by the activity of zinc family member 2 (ZIC2). This leads to the promotion of EC cell proliferation and migration by activating the Notch signalling pathway. On the other hand, SNGH12 expression can be suppressed by miR-4429 [113,114]. On the other hand, the next small nucleolar host gene—SNHG14—acts as an obstructive factor in EC. In previous research, it was proved that the overexpression of SNGH14 hampered the migration, invasion, and viability of endometrial cancer cells. This lncRNA inhibits miR-93-5p and upregulates zinc finger and BTB domain-containing protein 7A (ZBTB7A)—a factor that acts a tumour suppressor [115]. However, the role of SNHG14 in endometrial cancer is not entirely clear. According to other research, SNHG14 can promote the proliferation of EC cells. High expression of SNHG14 was correlated with a larger tumour size, shorter overall survival, and a more advanced pathological stage. The expression of SNHG14 was downregulated by miR-655-3P [116]. Other SNHGs—16 and 25—promote the proliferation and malignancy of EC. SNHG16 stimulates hexokinase 2 (HX2) expression by blocking miR-490-3p. HX2 increases glycolysis and endometrial carcinoma proliferation [117]. Higher expression of SNHG25 is correlated with shorter overall survival rates among patients with EC. SNHG25 upregulates the level of fatty acid synthase (FASN), which plays the role of an oncogene and impedes the expression of microRNA-497-5p [118]. The main characteristics of small nucleolar host genes are summarized in Table 5.

### 4.8. LncRNA-Associated Transcripts (ATs)

LncRNA-associated transcripts are a group of lncRNAs that are correlated with several types of tumours. The example of this kind of lncRNA is colon cancer-associated transcript 1 (CCAT1). CCAT1 acts as a promotor of EC progression, and it stimulates proliferation and migration. This activity is correlated with the downregulation of miR-181a-5p [119]. In other research, CCAT1 expression was 9.3-fold higher in endometrial cancer compared to normal endometrium, and the inhibition of this lncRNA expression was correlated with a significant proliferation reduction [120]. Another lncRNA AT that is engaged with the EC course is prostate cancer-associated transcript 1 (PCAT1). This lncRNA expression is positively correlated with myometrial invasion, the FIGO stage, lymph node metastasis, and a shorter overall survival. Overexpression of PCAT1 is relevant to changes in the molecular level of different molecules [121]. PCAT1 promotes the downregulation of E-cadherin in correlation with EZH2. E-cadherin downregulation is associated with an increase in the epithelial–mesenchymal transition (EMT). It stimulates the progression and metastasis of EC [122]. Another lncRNA that promotes EC development is retinoblastoma-associated transcript-1 (RBAT1). This molecule increases cell viability and suppresses cell apoptosis. Moreover, RBAT1 reduces the chemosensitivity of EC cells to carboplatin/paclitaxel. A potential mechanism is correlated with promoting the expression of multidrug-resistant-related protein and the inhibition of miR-27b [123]. The last LncRNA in this group is bladder cancer-associated transcript 2 (BLACAT2). High expression of this molecule is correlated with migration and invasion of EC cells. It also promotes proliferation. BLACAT2 suppresses miR-378a-3p activity, and as a result, it induces mitogen-activated protein kinase/extracellular signal-regulated kinase (MEK/ERK) pathway. It is correlated with tumour development [124]. The main characteristics of lncRNA-associated transcripts are summarized in Table 6.

### 4.9. Other Long Non-Coding RNAs Associated with Endometrial Cancer

In addition to the above, there are also other lncRNAs that influence the course of endometrial cancer, and that cannot be assigned to a specific group. Some of them stimulate tumour development, while others may inhibit it. The number of lncRNAs among which associations with EC are being discovered is constantly growing.

### 4.10. LncRNAs That Promote Endometrial Cancer Progression

Another group of lncRNAs includes a variety of molecules. Most of them stimulate the development of endometrial cancer. LncRNA deleted in lymphocytic leukaemia 1 (DLEU1) is an example of a factor which promotes EC development. DLEU1 increases viability, migration, and invasion, and reduces the proportion of apoptosis. It is correlated with an increase in PI3K/AKT/mTOR pathway expression. DLEU1 also decreases the activity of miR-490 [125,126]. The DLEU1 level can be considered as an independent prognostic factor in EC [127]. Another example of a molecule from this group is DLEU2. It can promote EC progression by upregulating hexokinase 2 (HK2) expression. It leads to intensified glycolysis and epithelial–mesenchymal transition [128]. Another lncRNA whose high expression is correlated with a high FIGO stage and poor tumour differentiation is LncRNA-ATB. It is inhibited by miR-126 [129]. Furthermore, lncRNA that promotes EC development by the upregulation of the proliferation of cells is BANCR. The expression of this molecule is positively correlated with ERK/MAPK signalling pathway activation and the production of MMP2 and MMP1. These molecules stimulate invasion and metastasis [130]. Moreover, lncRNA PVT1 stimulates cell proliferation, migration, and invasion, and blocks apoptosis. PVT1 inhibits the miR-195-5p activity. It causes the upregulation of acidic fibroblast growth factor receptor (FGFR1) and basic fibroblast growth factor (FGF2). Both factors activate the PI3K/AKT and MAPK/Erk pathways, which promote EC progression. PVT1 can also stimulate the expression of Centromere Protein H (CENP-H) by impeding miR-612 [131,132]. LncRNA XIST participates in EC growth by stimulating the expression of Centromere coiled-coil protein 110 (CCP110), a protein that stimulates migration, invasion, and proliferation. XIST blocks the activity of miR-129-2-3p, a factor that impedes CCP110 expression [133]. Another lncRNA is NDRG1. It promotes viability, migration, and invasion. NDRG1 stimulates neovascularization by upregulating the expression of VEGFA [134]. The next lncRNA that upregulates the epithelial–mesenchymal transition is MIR210HG. It is correlated with tumorigenesis, metastasis, and drug resistance. This lncRNA inhibits miR-337-3p and promotes the Wnt/β-catenin and TGF-β/Smad3 signalling pathways. These pathways engage in the epithelial–mesenchymal transition [135]. LSINCT5 also activates the Wnt/β-catenin pathway by stabilization of High-mobility group AT-hook 2 (HMGA2) [136]. Another lncRNA is ROR, which promotes the expression of the Notch1 protein, and causes the upregulation of proliferation and downregulation of apoptosis in EC [137]. LncRNA FIRRE promotes endometrial cancer development and reduces radiotherapy sensitivity. FIRRE downregulates the expression of miR-199b-5p. This microRNA blocks the activity of sirtuin 1 (SIRT1), which takes part in the FIRRE-mediated stimulation of the autophagy process. Moreover, FIRRE increases the viability and proliferation of EC cells [138]. Another lncRNA molecule which promotes cell proliferation, migration, and invasion in endometrial cancer is LINP1. It can activate PI3K/AKT signalling and promotes tumour invasion [139]. In another study, it was proved that the high expression of lncRNA RNA (THOR) was correlated with a poor overall survival rate. THOR promotes the proliferation, invasion, and migration of EC cells. It is in relation to the upregulation of the ERK/AKT pathway [140]. Furthermore, PCGEM1 is upregulated in endometrial cancer. This lncRNA acts as a tumour development promotor. PCGEM1 inhibits the expression of miR-129-5p, as well as stimulates STAT3 activity [141]. Another lncRNA that participates in EC growth via the MAPK pathway activation is HEIH. It leads to the promotion of chemo-resistance in endometrial cancer cells and enhances cell proliferation and viability [142]. Another lncRNA that promotes EC growth via increasing the epithelial–mesenchymal transition is SLERT. It elevates vimentin and N-cadherin expression and downregulates E-cadherin [143]. TUG1 demonstrates a similar mechanism of action, and it also favours the epithelial–mesenchymal transition due to increasing the levels of E-cadherin and a decrease in N-cadherin. The overexpression of this molecule is correlated with lower survival rates among patients with EC and with lymph node metastasis [144]. TUG1 acts as a suppressor of miR-299 and miR-34a-5p expression [145]. The last one lncRNA to mention here has a diverse impact on EC development. It is UCA1. It promotes cell proliferation, colony formation, invasion, and lymph node metastasis, and inhibits apoptosis. UCA1 overexpression is correlated with a higher malignancy of EC. A decrease in UCA1 expression leads to the suppression of tumour growth and cancer progression. This lncRNA inhibits miR-204-5p and promotes Kruppel-like factor 5 (KLF5) and relaxin-like family peptide receptor 1 (RXFP1). These factors participate in metastasis formation [146,147,148].

### 4.11. lncRNAs That Suppress Endometrial Cancer Progression

In addition to lncRNAs that stimulate the development of endometrial cancer, there are also those that block its growth. They act through various molecular mechanisms and influence the proliferation, migration, and invasion of cancer cells. The first lncRNA FER1L4 inhibits EC progression by the upregulation of PTEN and inhibition of Akt activation. It leads to a decline in proliferation, as well as an increase in apoptosis [149]. Lnc-NA stimulates the apoptosis of EC by activation of the nuclear receptor subfamily 4 group A member 1 (NR4A1) gene. It leads to the activation of the apoptosis signalling pathway to inhibit tumour progression [150]. Another LncRNA, MONC, impedes endometrial cancer tumour development, as well as migration and invasion. MONC decreases the expression of miR-636 and impedes the epithelial-to-mesenchymal transition process. The downregulation of miR-636 is correlated with the upregulation of Glucuronic Acid Epimerase (GLCE), which inhibits the progression of EC [151]. In another study, LncRNA TUSC7 inhibits cell proliferation, cycle progression, and metastasis. TUSC7 acts as a suppressor of miR-616, which is a factor that blocks the expression of suppressor of cytokine signalling 4 (SOCS4). SOCS4 inhibits the angiogenesis and development of tumour. Moreover, TUSC7 can block an activity of miR-23b, and as a result, it increases the sensitivity to chemotherapy based on cisplatin and paclitaxel [64,152,153]. Another lncRNA that impedes EC progression is NORAD. The downregulation of this molecule is correlated with EC progression (FIGO stage) and poor outcome. NORAD stimulates the activity of pro-apoptotic Far Upstream Element-Binding Protein 1 (FUBP1). It leads to the apoptosis of EC cells [154]. The next lncRNA that inhibits EC development is ZXF1. It impedes the migration and invasion of EC, as well as regulates cell cycle and proliferation. ZXF1 can stimulate the expression of cyclin-dependent kinase inhibitor 1A (P21) via inhibiting the miR-378a-3p- and CDC20-mediated degradation of P21. P21 participates in cell cycle arrest and the stimulation of cell apoptosis [155]. Another lncRNA, FENDDR, suppresses cell proliferation and stimulates apoptosis by the downregulation of SOX4 expression [156]. LncRNA can also block EC cells in the G1 phase. An example of this is XLEC1. It inhibits the proliferation and migration of EC cells. XLEC1 blocks cells in the G1 phase via Myc Promoter-Binding Protein-1 (MBP-1) activation and the suppression of c-Myc expression. C-myc acts as an oncogene that participates in cellular growth [157]. Furthermore, lncRNA GAS5 functions as a tumour development suppressor by the upregulation of cancer cell apoptosis. GAS5 enhances the expression of PTEN via the suppression of miR-103 activity. Moreover, GAS5 inhibits EC progression by regulating the immune system. GAS5 promotes phagocytosis, antigen presentation, and the activation of cytotoxic T cells. It changes the phenotype of tumour-associated macrophages from a pro-tumour to an anti-tumour [158,159]. The last lncRNA to mention that suppresses EC growth is lncRNA-LA16C-313D11.11. It also acts as a promotor of PTEN expression. LA16C-313D11.11 inhibits the activity of microRNA-205-5p—a factor that blocks PTEN [160]. The main characteristics of other lncRNAs are summarized in Table 7.

## 5. Conclusions

In summary, various lncRNAs may influence the course of endometrial cancer. Most of them stimulate tumour growth by increasing proliferation, migration, and invasion while reducing apoptosis. However, some of them inhibit the progression of endometrial cancer by blocking the cell cycle, proliferation, migration, and invasion due to blocking the epithelial–mesenchymal transition. They can also induce the apoptosis of cancer cells. LncRNAs exert their effects through a variety of molecular mechanisms. They stimulate or inhibit the activity of molecular pathways, proteins, and enzymes. They also influence the expression of several microRNA molecules. In the future, selected lncRNAs may be used as prognostic factors determining the clinical course of the disease, the degree of cancer malignancy, and the prognosis for patients. They may also be considered when selecting the optimal therapy. Some of them can become direct therapeutic targets. The influence of drugs on the inhibition or stimulation of lncRNAs may affect the course of the disease and improve the prognosis and overall survival rate. LncRNAs are a part of the modern approach to the development and a course of cancer diseases as a set of genetic and molecular disorders encompassing a number of factors. Further research should focus on isolating panels of lncRNAs or single molecules that would constitute prognostic factors for the course of endometrial cancer. We should also look for drugs that, by acting on selected long non-coding RNAs, would inhibit the progression or recurrence of the cancer. The literature data indicate that changing the expression of selected lncRNAs may be important for tumour development, invasion, and metastasis. Potential therapeutic targets among lncRNAs may include, in particular, molecules such as MEG3 or HOTAIR. This is related to their comprehensive impact on oncogenesis, studied in many cancers. The first research on the use of these molecules is currently being conducted.

## Figures and Tables

**Table 1 cancers-16-02125-t001:** Main characteristics of subtypes of lncRNA.

Type of lncRNA	Characteristic Features
Intronic lncRNA	Located in intronic region of protein-coding gene
Intergenic lncRNA	Situated between two protein-coding genes
Antisense lncRNA	Transcription from complementary DNA strands
Bidirectional lncRNA	Arise from bidirectional transcription of protein-coding genes
Enhancer lncRNA	Originate from enhancer regions of protein-coding gene promoters; involved in the mediation of transcription factor positioning
Circular lncRNA	lncRNA that undergo back splicing

**Table 2 cancers-16-02125-t002:** Summary of biological functions of lncRNAs.

Biological Functions of lncRNAs
Regulation of chromatin structure
Alteration of DNA methylation
Influence transcription process
Post-transcriptional regulation
Stimulation of DNA/RNA decay
Ensure stability of mRNA
Encoding peptides and protein relocalization
Regulation of translation and post-translation modification

**Table 3 cancers-16-02125-t003:** Summary of long non-coding antisense RNA actions.

Long Non-Coding Antisense RNA		
Name of lncRNA	Impact on Endometrial Cancer Course	Molecular Target of lncRNA Action
NR2F1-AS1	Promoting progression	SOX4
LOXL1		RAP1B
MCM3AP-AS1		VEGF
AFAP1-AS1		VEGFA
NNT-AS1		Estrogen
DLX6-AS1		p300/E2F1 acetyltransferase
TTN-AS1		PUM2
GATA3-AS1		ARRB2
HOXB-AS3		ADAM9; lipid metabolism through PTBP1
HOXA-AS2		ZFX
HOXC-AS2		HKDC1
RHPN1-AS1		MAPK/ERK pathway
ABHD11-AS1		Cyclin D1
DCST1-AS1		CADM1 and HOXB5
DSCAM-AS1		PRL and Erα
VPS9D1-AS1		BIRC5 and SGK1
BMPR1B-AS1		DCLK1/Akt/NF-κB pathway and ZFAS
TMPO-AS1		GLUT1 and paclitaxel resistance
CDKN2B-AS1		Paclitaxel resistance
FGD5-AS1		PD-1/PD-L1 checkpoint and paclitaxel resistance
FRMD6-AS2	Inhibiting progression	Hippo pathway
MTCP1-AS1		SMAD-7
EIF1AX-AS1		EIF1AX
SOCS2-AS1		AURKA
GATA6-AS		MMP-9
OIP5-AS1		PTEN/AKT pathway
HAND2-AS1		NMU

**Table 4 cancers-16-02125-t004:** Summary of long intergenic non-coding (LINC) RNA actions.

Long Intergenic Non-Coding RNA (LINC)		
Name of lncRNA	Impact on Endometrial Cancer Course	Molecular Target of lncRNA Action
Linc02936	Promoting progression	Ferroptosis
Linc01016		NFYA
Linc01410		CHD7
Linc01857		MYCN
Linc00470		MYC
Linc01194		SOX2
Linc01541		VEGFA
Linc00958		E2F3; PHF6; TCF-4
Linc01224 and LINC01170		AKT3
Linc01220		MAPK11
Linc00461		COX-2
Linc01106.		MET
Linc00672	Inhibiting progression	p-53-induced LASP1 downregulation; increasing chemosensitivity to paclitaxel
Linc00261		FOXO1
Linc00478		PTBP1
Linc01589		Immune system activation

**Table 5 cancers-16-02125-t005:** Summary of small nucleolar host gene (SNHG) actions.

Small Nucleolar Host Genes (SNHGs)		
Name of lncRNA	Impact on Endometrial Cancer Course	Molecular Target of lncRNA Action
SNHG9	Promoting progression	Glycolysis
SNHG12	Promoting progression	MMP2 and MMP9
SNHG14	Inhibiting progression	ZBTB7A
SNHG16	Promoting progression	HX2
SNHG25	Promoting progression	FASN

**Table 6 cancers-16-02125-t006:** Summary of long non-coding RNA-associated transcript actions.

LncRNA-Associated Transcripts (ATs)		
Name of lncRNA	Impact on Endometrial Cancer Course	Molecular Target of lncRNA Action
CCAT1	Promoting progression	miR-181a-5p
PCAT1		epithelial–mesenchymal transition
RBAT1		carboplatin/paclitaxel chemosensitivity downregulation
BLACAT2		MEK/ERK pathway

**Table 7 cancers-16-02125-t007:** Summary of other long non-coding RNA actions.

Other Long Non-Coding RNAs Associated with Endometrial Cancer		
Name of lncRNA	Impact on Endometrial Cancer Course	Molecular Target of lncRNA Action
HOTAIR	Promoting progression	PI3K/Akt pathway; PTEN suppression
H19		HOXA10
NEAT1		EZH2; STAT3; HMGA1; TIMD4; MMP-2, MMP-7 and IGF1
DLEU1		PI3K/AKT/mTOR pathway
DLEU2		HK2
LncRNA-ATB		ATB
BANCR		ERK/MAPK pathway; MMP2 and MMP1
PVT1		FGFR1 and FGF2; CENP-H
XIST		CCP110
NDRG1		VEGFA
MIR210HG		Wnt/β-catenin and TGF-β/Smad3 signalling pathways
LSINCT5		HMGA2
ROR		Notch1
FIRRE		SIRT1
LINP1		PI3K/AKT signalling
THOR		ERK/AKT pathway
PCGEM1		STAT3
HEIH		MAPK pathway
SLERT		Vimentin and E-cadherin upregulation; N-cadherin downregulation
TUG1		Epithelial–mesenchymal transition
UCA1		KLF5 and RXFP1
MEG3	Inhibiting progression	PD-L1; PI3K/m-TOR signalling; Notch 1 and HES-1 expression
FER1L4		PTEN
NA		NR4A1
MONC		GLCE
TUSC7		SOCS4
NORAD		FUBP1
ZXF1		P21
FENDDR		SOX4
XLEC1		MBP-1 activation and c-Myc downregulation
GAS5		PTEN
lncRNA-LA16C-313D11.11		PTEN

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
