# Peer review of "The Promotive and Inhibitory Role of Long Non-Coding RNAs in Endometrial Cancer Course—A Review"

_cancers, 2024, doi:10.3390/cancers16112125_

Round 1

Reviewer 1 Report

Comments and Suggestions for Authors

The article sent for review entitled: The role of long non-coding RNA in endometrial cancer – review is well situated on the current medicinal problem i.e.: early diagnosis. Endometrial cancer as correctly focused by authors is one of the most popular in the female population.  The author correctly discussed cancer development and style of life, smoking, obesity etc. Moreover, the authors mention the old therapeutic strategy i.e.: antisense. As a review article, it can be valuable for readers in the field. Moreover, it focuses on the importance of the level of LncRNA in cancer development, progression and in some cases suppression. The article is well-written and readable. Plenty of current articles have been correctly cited.

In conclusion, the article can be valuable for readers in my opinion the results and topic are well-fitted to the journal (Cancers) scope.

In the introduction and throughout the article, I hope that it is only a typing mistake:

It is LncRNAs coding genes it should be LncRNAs-coding genes otherwise it means that RNA is the genome.

Comments on the Quality of English Language

The article sent for review entitled: The role of long non-coding RNA in endometrial cancer – review is well situated on the current medicinal problem i.e.: early diagnosis. Endometrial cancer as correctly focused by authors is one of the most popular in the female population.  The author correctly discussed cancer development and style of life, smoking, obesity etc. Moreover, the authors mention the old therapeutic strategy i.e.: antisense. As a review article, it can be valuable for readers in the field. Moreover, it focuses on the importance of the level of LncRNA in cancer development, progression and in some cases suppression. The article is well-written and readable. Plenty of current articles have been correctly cited.

In conclusion, the article can be valuable for readers in my opinion the results and topic are well-fitted to the journal (Cancers) scope.

In the introduction and throughout the article, I hope that it is only a typing mistake:

It is LncRNAs coding genes it should be LncRNAs-coding genes otherwise it means that RNA is the genome.

Author Response

Dear Reviewer, 

Thank you for your time and your extremely valuable opinion. Indeed, LncRNAs coding genes was a typing mistake and we corrected it. Additionally, article has been checked for linguistic correctness by an expert translator.

Kind regards,

Patryk Jasielski

Reviewer 2 Report

Comments and Suggestions for Authors

This is a very well written review .

I appreciate the reviewers  explanation of  the various types of lncRNAs

The discussion of the topic materials was relevant. 

What would be most helpful is to have a figure describing the various functions of lncRNAs in a biological setting or ontological setting 

One question would be to determine if there is a hi, mid, low risk category, for the plots generated in figure 2,.  Part of the reason is that after 100 days, the number of patients on trial are in many cases less than 10  This could also explain why the HR rates are generally less than 2.0.   In figure 3, many cells lines that model lung tumors do not have any methylation of SORT1A.  Though I would imagine there are a number of regions where SORT1 could be methylated. Have the authors only looked at one generic region of Sort1A and b to determine methylation status?   IN figure 4, it is not quite clear wny the SORT 1 in normal PMBCs would be methylated ? Since we are assessing lung cancer, it would be difficult to use SORT1 methylation as a screening tool to find metastatic lung cancer cells in thousands of PMSCs.    In figure 5, the gender linkage to SORT1A/B methylation is unclear because is there any linkage or connection to lung cancer and gender.  Mostly increases in lung cancer rates in women were due to increase smoking rates in women in the 70s / 80s, not genetics

Comments on the Quality of English Language

 N/a

Author Response

Dear Reviewer, 

Thank you for your time and your extremely valuable opinion. 

What would be most helpful is to have a figure describing the various functions of lncRNAs in a biological setting or ontological setting 

we have added a table summarizing the biological functions of lncRNAs in the body under the section "Mechanisms of lncRNAs action:"

One question would be to determine if there is a hi, mid, low risk category, for the plots generated in figure 2,.  Part of the reason is that after 100 days, the number of patients on trial are in many cases less than 10  This could also explain why the HR rates are generally less than 2.0.   In figure 3, many cells lines that model lung tumors do not have any methylation of SORT1A.  Though I would imagine there are a number of regions where SORT1 could be methylated. Have the authors only looked at one generic region of Sort1A and b to determine methylation status?   IN figure 4, it is not quite clear wny the SORT 1 in normal PMBCs would be methylated ? Since we are assessing lung cancer, it would be difficult to use SORT1 methylation as a screening tool to find metastatic lung cancer cells in thousands of PMSCs.    In figure 5, the gender linkage to SORT1A/B methylation is unclear because is there any linkage or connection to lung cancer and gender.  Mostly increases in lung cancer rates in women were due to increase smoking rates in women in the 70s / 80s, not genetics

I would like to point out that this review has probably been attached to our manuscript by some kind of mistake. Our article is a review on the role of lncRNAs in endometrial cancer. It is not a research paper, so it does not contain any results or figures.

Moderate editing of English language required

Our text was checked by a translator who corrected any linguistic errors.

Thank you once again for Your precious time.

Kind regards,

Patryk Jasielski 

Reviewer 3 Report

Comments and Suggestions for Authors

Patryk Jasielski and co-authors present a high quality and well-written review manuscript focused on the role of long non-coding RNA in endometrial cancer.

Authors suggest that endometrial cancer is one of the most common malignant tumours in women. Its incidence is increasing, especially in developed countries. Development of this tumour is associated with several genetic disorders, many of which are still unknown. Their discover may be important in optimizing therapy, but also in determining the prognosis, the risk of disease recurrence or the formation of metastases. One type of RNA molecules currently being intensively studied in many types of cancer are long non-coding RNAs (lncRNAs). LncRNAs coding genes occupy a large fraction of the human genome. LncRNAs regulate many aspects of cell development, metabolism, and other physiological processes. Research revealed that they participate in cancer development either. 

Authors claim that diverse types of lncRNA can function as a tumour-suppressor or an oncogene that can alter migration, invasion, cell proliferation, apoptosis, and immune system response. Hence in the future lncRNA can be useful molecular marker for early diagnosis of cancer and a new target for therapy. Recent studies suggest that selected lncRNAs are important in endometrial cancer course.

Authors describe over 70 lncRNA involved in the development of endometrial cancer, which were studied in vivo and in vitro research. The aim of this study is to review the role of lncRNA in development of carcinoma of uterus body.

Authors cover such aspects as:

- Mechanisms of lncRNAs action

- The role of lncRNA in endometrial cancer

- Long non-coding antisense RNAs (-AS)

- Long intergenic non-coding RNA (LINC)

- Maternally expressed gene 3 (MEG3)

- Homeobox (HOX) transcript antisense intergenic RNA (HOTAIR)

- Nuclear enriched abundant transcript 1 (NEAT1)

- Small Nucleolar Host Genes (SNHG)

- LncRNAs associated transcripts (AT)

- Other long non-coding RNAs associated with endometrial cancer

- LncRNAs that promote endometrial cancer progression

- lncRNAs that suppress endometrial cancer progression

Finally, authors conclude that various lncRNAs may influence the course of endometrial cancer. Most of them stimulate tumour growth by increasing proliferation, migration and invasion while reducing apoptosis. However, some of them inhibit the progression of endometrial cancer by blocking the cell cycle, proliferation, migration, and invasion because of blocking the epithelial-mesenchymal transition. 

Overall, the manuscript is highly valuable for the scientific community and should be accepted for publication.

======================

Other comments to authors:

1) Please check for typos throughout the manuscript.

2) Please improve figures/tables where appropriate.

3) Lines 239, 527,  Table 2, Table 6. With regards to SOCS proteins – authors are kindly encouraged to cite the following article that describes important regulatory properties of SOCS protein (might be relevant for endometrial cancer). DOI: 10.1371/journal.pone.0131218

Author Response

Dear Reviewer,

Thank you for your time and your extremely valuable opinion. 

1) Please check for typos throughout the manuscript.

2) Please improve figures/tables where appropriate.

We checked typos and made improvements to figures/tables where appropriate.

3) Lines 239, 527,  Table 2, Table 6. With regards to SOCS proteins – authors are kindly encouraged to cite the following article that describes important regulatory properties of SOCS protein (might be relevant for endometrial cancer). DOI: 10.1371/journal.pone.0131218

Indeed, it is valuable manuscript for our article and we added it to our references. 

Thank you again for your time.

Kind regards,

Patryk Jasielski  

Reviewer 4 Report

Comments and Suggestions for Authors

The manuscript offers a comprehensive overview of the current research surrounding the role of various long non-coding RNAs (lncRNAs) in the development, progression, and potential treatment of endometrial cancer. It provides a detailed examination of both the inhibitory and promotive effects of lncRNAs through multiple molecular mechanisms. While the scope and depth of the review are commendable, there are several areas where clarity and coherence can be improved to enhance the paper's impact and readability. My recommendation on this paper is major revision.

1. Consider refining the title to specify the focus on both promotive and inhibitory roles of lncRNAs in endometrial cancer to immediately convey the dual aspects of your study's focus to the reader.

2. The abstract should concisely summarize the key points of the review, emphasizing the novel insights your work brings to the field of endometrial cancer research. Avoiding excessive detail will enhance clarity and impact.

3. The introduction would benefit from a clearer exposition of the problem statement and a more explicit articulation of the research gap your review intends to fill. This will provide readers with a better understanding of the context and significance of the review.

4. In the literature review section, ensure that each paragraph transitions smoothly into the next, maintaining a thematic coherence that is currently lacking in some parts. This will help in building a persuasive narrative.

5. Clarify the criteria used for selecting studies included in the review. A detailed methodology enhances the reproducibility of the review and provides transparency about how conclusions were drawn.

6. Ensure consistency in the use of terms and definitions throughout the manuscript. For example, if you are using different terms interchangeably (e.g., "lncRNAs" and "long non-coding RNAs"), clarify this early in the document.

7. Some tables are dense with information, which could be overwhelming for readers. Consider simplifying these or providing additional commentary to guide readers through the data.

8. The conclusions could be strengthened by succinctly summarizing the implications of your findings for clinical practice or future research. Highlighting the potential therapeutic targets among lncRNAs could be particularly impactful.

Comments on the Quality of English Language

Moderate editing of English language required.

Author Response

Dear Reviewer,

Thank you for your time and your extremely valuable opinion. We have analyzed your instructions and made some corrections to our article.

  1. Consider refining the title to specify the focus on both promotive and inhibitory roles of lncRNAs in endometrial cancer to immediately convey the dual aspects of your study's focus to the reader.

We changed the title of the article to "The promotive and inhibitory role of long non-coding RNA in endometrial cancer course – review."

  1. The abstract should concisely summarize the key points of the review, emphasizing the novel insights your work brings to the field of endometrial cancer research. Avoiding excessive detail will enhance clarity and impact.

The abstract has been modified and shortened.

  1. The introduction would benefit from a clearer exposition of the problem statement and a more explicit articulation of the research gap your review intends to fill. This will provide readers with a better understanding of the context and significance of the review.

We have added to the end of introduction a few information: “Due to the growing number of endometrial cancer cases the search for individualized therapy that is less burdensome for patients is required. LncRNAs have broad and incompletely described effects on a number of metabolic processes. Due to the lack of a similar article, the topic of our review is the comprehensive description of the stimulatory and inhibitory effects of lncRNAs on the course of endometrial cancer. we took into account the effect of lncRNA both stimulating and inhibiting the progression of endometrial cancer, because in the future there is a chance to use drugs with different mechanisms of action.”

  1. In the literature review section, ensure that each paragraph transitions smoothly into the next, maintaining a thematic coherence that is currently lacking in some parts. This will help in building a persuasive narrative.

We have modified the content so that subsequent paragraphs flow smoothly from one to the next.

  1. Clarify the criteria used for selecting studies included in the review. A detailed methodology enhances the reproducibility of the review and provides transparency about how conclusions were drawn.

A materials and methods chapter has been added, where the methodology used in our article is described.

  1. Ensure consistency in the use of terms and definitions throughout the manuscript. For example, if you are using different terms interchangeably (e.g., "lncRNAs" and "long non-coding RNAs"), clarify this early in the document.

Consistency in the use of terms and definitions throughout the manuscript has been corrected.

  1. Some tables are dense with information, which could be overwhelming for readers. Consider simplifying these or providing additional commentary to guide readers through the data.

The content of the tables has been simplified.

  1. The conclusions could be strengthened by succinctly summarizing the implications of your findings for clinical practice or future research. Highlighting the potential therapeutic targets among lncRNAs could be particularly impactful.

We have added a brief summary of our results and highlighted two lncRNAs - MEG3 and HOT-AIR as potential therapeutic targets.

Moderate editing of English language required

Our text was checked by a translator who corrected any linguistic errors.

Thank you once again for Your precious time.

Kind regards,

Patryk Jasielski

Round 2

Reviewer 4 Report

Comments and Suggestions for Authors

Upon reviewing the revised manuscript (v2) of "The role of long non-coding RNA in endometrial cancer," several notable improvements were observed in response to the feedback provided on version 1. Nonetheless, although the literature review section shows improved thematic coherence, some sections still require further streamlining. For instance, the transitions between the descriptions of different lncRNAs could be made smoother to ensure a more cohesive narrative flow, which would aid in maintaining the reader’s engagement and comprehension.

Comments on the Quality of English Language

Moderate editing of English language required.

Author Response

Dear reviewer,

We would like to thank you for your time and useful tips.

Upon reviewing the revised manuscript (v2) of "The role of long non-coding RNA in endometrial cancer," several notable improvements were observed in response to the feedback provided on version 1. Nonetheless, although the literature review section shows improved thematic coherence, some sections still require further streamlining. For instance, the transitions between the descriptions of different lncRNAs could be made smoother to ensure a more cohesive narrative flow, which would aid in maintaining the reader’s engagement and comprehension.

We checked once again the literature review section and improved thematic coherence.

Moderate editing of English language required.

Our text has been checked by a qualified translator - Ms. Magdalena Rejman-Zientek. It is her e-mail adress: mzientek@prz.edu.pl.

Kind regards,

Patryk Jasielski